# An Observational Real-Life Study with a New Infant Formula in Infants with Functional Gastro-Intestinal Disorders

**DOI:** 10.3390/nu13103336

**Published:** 2021-09-23

**Authors:** Yvan Vandenplas, Laetitia Gerlier, Karin Caekelbergh, Mike Possner

**Affiliations:** 1KidZ Health Castle, Vrije Universiteit Brussel (VUB), UZ Brussel, 1090 Brussels, Belgium; 2IQVIA, 1930 Zaventem, Belgium; Laetitia.Gerlier@iqvia.com (L.G.); Karin.Caekelbergh@iqvia.com (K.C.); 3Nestlé Nutrition Institute, 60528 Frankfurt am Main, Germany; Mike.Possner@de.nestle.com

**Keywords:** constipation, colic, CoMiSS, crying, functional gastro-intestinal disorder, gastro-esophageal reflux, regurgitation, QUALIN, quality of life

## Abstract

Functional gastro-intestinal disorders (FGIDs) impair the quality of life of many infants and their families. A formula with partial whey hydrolysate, starch, high magnesium content, prebiotic fructo-oligosaccharide and galacto-oligosaccharide and the probiotic *Lactobacillus reuteri* DSM 17938 was given during two weeks to 196 infants with at least two FGIDs. The efficacy was evaluated with the Cow Milk-associated Symptom Score (CoMiSS^®^) and quality of life with the QUALIN score. The formula was shown to decrease FGIDs within three days (decrease of CoMiSS −1.29 (3.15) (mean (SD), *p* < 0.0001) followed by an improvement of quality of life after seven days (increase QUALIN +1.4 (7.8); *p*: 0.008). Constipation decreased from 18.8% to 6.5% within three days. In combination with reassurance and guidance, the nutritional intervention was shown to be effective in infants with FGIDS in real-life circumstances.

## 1. Introduction

Functional gastro-intestinal disorders (FGIDs) occur in more than 25% of all infants and are a frequent reason for parents to consult a health care provider [1,2,3,4]. Regurgitation, infantile colic and functional constipation are the most common FGIDs [1,2,3,4]. Although FGIDs have per definition no identifiable underlying organic cause [5], infants with a FGID and their families display a reduced quality of life (QoL) and consult more often than asymptomatic controls [6]. FGIDs are a frequent reason for hospitalization, parental anxiety and depression, loss of parental working days with relevant social consequences and/or administration of drugs [2,7]. Many infants present with a combination of several FGIDs, although the reported range varies from 4.2% to 77% [1,2,3,8].

Anticipatory guidance, reassurance and helping caregivers to cope with the infant’s symptoms and providing support for the infant–family interaction is the cornerstone of the management of FGIDs. However, the majority of parents expect “more”: they expect the health care provider to “do” something and to deliver a prescription. The spiral of increasing prescriptions of proton pump inhibitors and other medications testifies to this reality [8,9]. Therefore, nutritional treatment is an attractive option since it is safe, devoid of adverse effects and supported by evidence of benefit [10].

The aim of this open real-life observational intervention study was to evaluate in infants presenting with a least two FGIDs the efficacy of an infant formula containing five components for which there is evidence of benefit from randomized controlled trials: partial whey hydrolysate (pHF-W), *Limosilactobacillus* (L.) *reuteri* DSM 17938, a prebiotic mixture of galacto-oligosaccharides (GOS) and fructo-oligosaccharides (FOS), high magnesium and potato starch.

## 2. Materials and Methods

The study design was a prospective, multicenter, observational study over 14 days with the study formula (Table 1: composition of study formula) in full formula-fed infants 0–4 months old presenting with at least two FGIDs out of regurgitation, constipation and crying.

Eighty-two pediatricians agreed to participate in the study. The parents of potential eligible infants were invited to participate in the study by the pediatrician. Enrollment was proposed when the parents consulted because of FGID symptoms. In order to limit selection bias, all consecutive infants fulfilling eligibility criteria for whom the parents provided consent were included.

Inclusion was Day 0; on Day 3 and 7 parents filled in a diary, and the closing visit was planned on Day 14 + 2. The trial was approved by the Ethical Committee of the UZ Brussel (B.U.N. 143202042971). Both parents signed the informed consent.

Since this was a real-world observational trial, infants did not have to fulfill Rome IV criteria to be eligible for inclusion. The Cow Milk-associated Symptom Score (CoMiSS^®^; Table 2) was used to quantify the severity of the symptoms. The CoMiSS was developed as a symptom score to quantify the symptoms in a trial comparing the efficacy of two extensive hydrolysates in infants suspected to suffer cow’s milk protein allergy (CMPA [11,12]. The CoMiSS ranges from 0 to 33.

Inclusion criterion was that the sum of the CoMiSS score for minimum two symptoms out of the three symptoms—crying, regurgitation, stools—needed to be >4, since 4 was reported to be the median value in presumed healthy infants [13]. Exclusion criteria were any food supplement except vitamins at inclusion, any previous use of a food for special medical purpose, current treatment with laxatives or antibiotics, suspected cow’s milk allergy. In other words: the study population consisted of presumed healthy formula-fed infants, except for the presenting manifestations of FGIDs, which were the reasons of the consultation. If the infant was included, the intervention consisted of 14 days exclusive feeding with the test formula.

QoL was evaluated with the QUALIN questionnaire [14]. This questionnaire was specially developed for use in infants and toddlers. The questionnaire includes 34 items with 6 possible answers, scored from −2 (quite false) to +2 (entirely true). The six possible answers were: false, mostly false, true and false, mostly true, true and do not know. The overall score ranges from −68 (poor QoL) and +68 (excellent QoL). Four topics are addressed: behavior and communication, ability to remain alone, family environment, and psychological and somatic well-being.

The primary endpoint was alleviation of symptoms according to the evolution of the CoMiSS and improvement of quality of life (QoL). Statistical analysis was done with SAS version 9.4 (SAS Institute Inc., Cary, NC, USA). Statistical significance was set at a level of 0.05. Standard deviation and interquartile ranges are reported for descriptive statistics. ANOVA and Student *t*-test were used to compare means of independent continuous variables, and when valid, a parametric paired Student *t*-test or a non-parametric Wilcoxon test was used to analyze changes from baseline in continuous outcomes between Day 0 and 14. The normality of the QUALIN and CoMiSS scores was assessed through the Shapiro–Wilk Test. Normality was assumed if the associated *p*-value of the test statistic was >0.05. Multivariate mixed effect model analyses were planned to assess change from baseline score over time (repeated measures). Multivariate mixed effect model was accommodated within-subject (patient in this case) non-independence or within-unit clustering (physician in this case) and unobserved heterogeneity, at once and allowed for subject-specific conclusions. This was controlled by introducing a random effect (patient) in the model [15]. Time (visit day 1, home report day 3, home report day 7, visit day 14) was also included as a discrete variable and fixed effect since the main interest was the change in mean scores [16]. Other covariates or factors (i.e., age, sex, BMI, gestational age, birth BMI, presence of adverse events, history of allergy, mode of delivery, number of siblings, baseline CoMiSS, baseline QUALIN, type of FGID at baseline) were also included in the model as fixed effects. The inclusion of baseline score as a covariate can help to decrease variability in random errors and increase the probability of detecting the significant time effect on the mean change from baseline in score [16]. Different models were constructed starting from the full model from which covariates are removed sequentially. The selection of the best model was based on the Akaike Information Criterion (AIC). The model with the lowest AIC was selected for the multivariate analysis. The normality of the QUALIN and CoMiSS scores was visually assessed through frequency distributions showing the spread of the scores. Normality was assumed if the distribution was bell-shaped and symmetric around the mean. Quantile–quantile (Q-Q) plots were constructed to assess whether the choice of the distribution was acceptable. Dots perfectly aligned with the straight line mean perfect choice of distribution. The “proc glimmix” procedure in SAS was used for the multivariate mixed effect model analyses in which a Gamma distribution and log link function was selected to model non-normal responses, otherwise a normal distribution was specified. Standard errors and two-sided 95% confidence intervals (CIs) are reported for the covariate estimates of the multivariate analyses (see Appendix A).

Secondary endpoints were the evolution of CoMiSS and QUALIN by subcategory, dietary changes, medications administered and adverse events. Analyses were reported for the full analysis set (FAS).

## 3. Results

Between May and December 2020, 196 infants were included by 51 (of the 82 who agreed to participate) pediatricians (number of inclusions/pediatrician: mean: 3.8; median: 2.0; Q1–Q3: (2.0–4.0). All data were available on Day 14 for 171 infants (87%). Reasons for the 13% missing data: not completed 3.6% (some data missing); lost in follow-up 3.6%; formula stopped: 5.6% (day 3: n = 1; day 7: n = 7; day 14: n = 3) (patient characteristics: Table 3). At baseline, all infants were full formula fed, but 92 (46.9%) of the infants had been breastfed during a short time (mean: 29.8 days; median: 21.0 days). The sum of the prevalence of the FGIDs was 218%, indicating that all included infants presented with at least two FGIDs and 18% with a combination of all three (crying, regurgitation, stool problems). Only a minority of infants had received drug treatment prior to inclusion: antibiotics (2.0%), antifungals (1.5%), acid blocking drugs (1.0%), analgesics (0.5%) and “OTC” products for digestive problems (3.6%). Atopic dermatitis was reported in 18 infants (9.2%) and the presence of respiratory symptoms in 30 (15.3%), indicating that the impact of these symptoms on the total CoMiSS was minimal.

The primary outcome, the evolution of the global CoMiSS, showed a statistically significant decrease of the CoMISS as soon as from Day 3 onwards, which was confirmed at Day 7 and 14 (Table 4). The mean decrease of baseline CoMiSS was 34%. Table 5 shows the evolution of the relevant symptoms in the CoMiSS and confirms that the scores for respiratory symptoms and skin manifestations were low (0.18, CoMiSS range 0–3; 0.20, CoMiSS range 0–12, respectively). The CoMiSS for stool consistency did not change because soft and fluid stools score, respectively, 2 and 4, while hard stools also score 4 (CoMiSS range 0–6). Therefore, Table 6 provides more detailed information on the evolution of stool consistency. Table 7 summarizes the change in CoMiSS in the subgroup with a CoMiSS > 9 (n = 44). Overall, efficacy of the dietary intervention was not different in the group with CoMiSS ≤ 9 and >9.

At Day 3, improvement of QUALIN was not yet significant (*p* = 0.065), but the change was significant from Day 7 onwards and continued to improve over time (Table 8).

Regarding CoMiSS, a multivariate analysis using a gamma distribution and log link function with the covariates CoMiSS baseline, age at inclusion, sex, weight at birth and inclusion, gestational age, mode of delivery, adverse events, duration symptoms showed that three factors were significantly and independently associated with the CoMiSS: baseline CoMiSS (*p* < 0.0001), mode of delivery (*p* = 0.012) and duration of symptoms (*p* = 0.005). Regarding QUALIN, the multivariate analysis using a normal distribution and the same covariates as for CoMiSS adding crying, showed that the following factors were significantly and independently associated: baseline QUALIN (*p* < 0.0001), age at baseline (*p* = 0.0008), weight al baseline (*p* = 0.0413) and duration of symptoms (*p* < 0.0001). The estimates of the covariates and diagnostic plots for CoMiSS and QUALIN scores can be found in the Appendix A.

During the two-week study period, gastro-intestinal drugs (alginate or proton pump inhibitors) were prescribed to 30 infants. The CoMiSS at baseline did not differ in the subgroups with or without GI drugs: 6.42 vs. 6.60, respectively. No differences in QUALIN were observed: in the subgroup without GI drugs, the evolution of QUALIN was from 22.67 to 25.80 (+3.11) and in the subgroup with GI medications, QUALIN increased from 23.90 to 26.25 (+3.18). In the subgroup not receiving GI drugs, the mean (SD) and median (Q1–Q3) change in CoMiSS over the study period was −1.54 (3.56) (*p* < 0.0001 to baseline) and −5 (−8; −1), respectively. The change in the subgroup receiving GI drugs was smaller: −1.64 (4.29) (*p* = 0.05) and −1 (−3.5; 0.5), respectively.

## 4. Discussion

This open interventional trial with a new therapeutic “comfort” formula containing pHF-W, GOS and FOS, the probiotic *L reuteri* DSM 17938, a high content of magnesium and starch was shown to decrease infant crying, regurgitation and constipation within three days (according to CoMiSS), subsequently increasing quality of life (according to QUALIN score). The rapid improvement of FGIDs in infants in these real-world study conditions is key in the management of FGIDs since improvement of symptoms will reassure parents.

The CoMiSS was developed as a symptom score in infants suspected to suffer cow’s milk protein allergy (CMPA) and was subsequently positioned as an awareness tool for CMPA [11,12]. The P95 cutoff in a healthy population was shown to be >9 [13]. Since regurgitation, infant distress, crying and stool consistency are, in combination with respiratory symptoms, atopic dermatitis and urticarial, the constituents of the CoMiSS, CoMiSS was used to show that respiratory and skin symptoms were virtually absent in this study population. The evolution of CoMiSS would then be used to show the evolution of the symptoms during the study intervention. The CoMiSS used the Bristol stool scale to describe stool consistency. Soft stools, type 5 according to Bristol, what is a normal consistency for infants, are accorded 2 points in the CoMiSS (Table 1). Furthermore, Bristol Type 6 (fluid stools) are normal in infants. Therefore, the contribution of the CoMiSS to describe the evolution of stool consistency is misleading. However, if interest is focused on the Bristol Type 1 and 2 (hard stools), a significant improvement of stool consistency could be demonstrated. It may be better to replace the Bristol stool scale by the Brussels Infant and Toddler Stool Scale in the CoMiSS [17]. The multivariate analysis showed that cesarean section and duration of symptoms were risk factors for the level of CoMiSS. Administration of GI drugs was not dependent on the level of CoMiSS or QUALIN (CoMiSS and QUALIN were slightly higher in the group without medication); neither was the evolution of both scores different in the groups with and without medication (the difference tended even to be slightly smaller in the group with medication). All these data confirm that GI drugs are not indicated in the management of FGIDs [10].

QUALIN score was independent of CoMiSS (< or >9), suggesting that at baseline QoL is more dependent on tolerance and bearing capacity of the caregivers than on severity of symptoms. However, when symptoms decreased, QUALIN increased (QoL improved). Thus, QoL is related to symptom severity but the impact of symptoms of QoL at baseline is subject to subjective interpretation.

According to literature data, it can be estimated that about 50% of infants present with a combination of FGIDs (range 4–77%) [1,2,3,8]. It might be difficult for the health care provider to discover the triggering mechanism. Therefore, a dietary approach with different components for which there is evidence for benefit from literature is an attractive approach. Moreover, dietary treatment is safe. At baseline, 29/196 (14.8%) infants were receiving a formula with the same protein content (pHF-W) as the test formula, but with standard magnesium content, without starch, without prebiotics or *L. reuteri*.

RCTs have previously shown efficacy for each of the specific ingredients: pHF-W, GOS and FOS, a high content of magnesium and starch. FOS and GOS are well-studied prebiotics in infant feeding. FOS and GOS have been known for the past 20 years to stimulate the growth of lactobacilli and bifidobacteria and to decrease possible pathogens in the GI microbiome [18,19,20,21,22].

GOS and FOS have additional benefits outside the management of FGIDs, such as protection for intestinal and extra-intestinal infections [23,24], resulting in decreased secretory IgA levels [25]. GOS and FOS offer also a possible indirect protection for allergy because of inducing a beneficial immunoglobulin profile in infants at high risk for allergy [26].

The pHF-W has been approved by the European Food Safety Association (EFSA) as a protein source suitable to be used in every infant [27,28]. A formula with a pHF-W protein is known to be nutritionally adequate [29,30]. pHF-W formulas are well accepted and tolerated [31]. The CoMiSS was developed as an awareness tool to consider a possible diagnosis of CMPA [12], and a cutoff > 9 was proposed to select a group of infants at risk to suffer CMPA [13]. In this study, 43/196 (21.9%) of the included infants had a CoMiSS > 9. However, the consulted pediatrician did not consider the diagnosis of CMPA in these infants, since suspicion of CMPA was an exclusion criterion and since pHF-W is not recommended in the management of CMPA [32]. However, it might be that some of the infants may suffer CMPA, since pHF-W protein does have some efficacy in the management of CMPA. In mice, pHF-W sensitization did not induce whey-induced clinical symptoms, even though sensitization was established [33]. Increased regulatory cell populations in the systemic immune system and a prevention of increased total Th1 and activated Th17 in the intestinal immune organs could contribute to the suppression of allergic symptoms [33]. An Italian study showed that 64% of infants with a positive double-blind challenge test with intact cow milk protein did tolerate a pHF-W formula [34]. A Japanese study showed oral tolerance in 1–9-year-old children with mild to moderate IgE-mediated CMA that 20 mL of cow milk was tolerated by 2/25, 20 mL of pHF-W by 16/25 and 20 mL of EHF by 22/25 [35]. These findings suggest that although pHF-W cannot be recommended in the management of CMPA, improvement or even disappearance of symptoms with a pHF-W does not exclude CMPA as a possible diagnosis. The question arises more and more of how to separate FGIDs from non-IgE-mediated mild to moderate CMPA in some infants, since symptoms and management do overlap. Could it just be considered as different wordings for the same condition?

This test formula does have a reduced lactose content (5.0 g/100 mL). Lactose is an important carbohydrate as it is the predominant carbohydrate in mother’s milk, which enhances the development of a GI microbiome rich in lactobacilli and bifidobacteria [36]. Lactose in infant formula improves calcium absorption [37]. However, a formula without lactose content has been shown to result in a clinically significant decrease of FGIDs [38,39]. Therefore, a reduced content of lactose may offer the balance between advantages and disadvantages of lactose.

### 4.1. Constipation: A Role for Magnesium, pHF-W, GOS and FOS

Magnesium-rich formula as a single change in the composition of infant formula was shown to be effective in the management of infant constipation, increasing the frequency and decreasing the hardness of the stools [40,41]. Compounds such as magnesium citrate work by pulling water into the intestines. The decrease in hardness of the stools is related to an increased water content [42]. Softer stools and increased frequency were directly related to a decrease of painful defecation, and thus decreased crying time [41]. An open trial with a pHF-W, GOS, *Bifidobacterium lactis* and high magnesium was reported to be effective in the management of constipation and improved many QoL aspects, such as sleep and work-related QoL, parent–child relationship, better social interaction with friends and relatives, resulting in a daily and overall improved QoL [43]. The test formula contains 12.49 mg magnesium per 100 kcal (8.37 mg/100 mL), which is about 50% more than the amount in a regular starter formula (8.5 mg/100 kcal). The European Delegated Act, EU DA 2016/127, recommends 5–15 mg/100 kcal.

There is evidence from RCTs that a formula with FOS and GOS, a pHF-W and starch is effective in the management of constipation and colic [44,45]. Defecation frequency in infants fed a pHF-W is almost twice the frequency of infants fed intact protein [46]. The prebiotic scGOS/lcFOS formulas have positive effects on stool characteristics such as stool consistency and stool frequency [47]. A formula with FOS and GOS, pHF-W and starch but also with the addition of a high concentration sn-2 palmitic acid resulted in a strong tendency of softer stools in constipated infants, but not in a difference in defecation frequency [48]. In an RCT, comparing a casein-dominant starter formula to a whey-predominant formula, with long-chain poly-unsaturated fatty acids and FOS and GOS, hard stools (0.7 vs. 7.5%, *p* < 0.001) were decreased with the test formula [49]. In comparison to the control group, the test group’s stool microbiota composition, gastric and intestinal transit times were closer to that of the breast-fed group [49].

The probiotic added to the test formula, *L. reuteri* DSM 17938, was also shown to reduce constipation in infants [50].

### 4.2. Colic: FOS and GOS, L. reuteri

The GI microbiome of infants presenting with colic is characterized by decreased numbers of lactobacilli and bifidobacteria, and more proteobacteria, including species producing gas and inflammation [51]. Since the FOS and GOS added to the test formula have been shown to stimulate the growth of a bifidogenic microbiome, the prebiotics could be beneficial in treating infant colic [44,45]. A fermented formula with FOS and GOS was shown to decrease colic [52].

The probiotic added to the test formula, *L. reuteri* DSM 17938, has been shown to be effective in the management of colic and several meta-analyses recommended its use in this indication [53,54].

### 4.3. Regurgitation: pHF, Probiotics and Starch

Carob bean gum and starch from corn and rice are the best studied thickeners and did not show a clinically relevant difference in efficacy [55]. The efficacy of potato starch to thicken infant formula has not been well studied, but potato starch is well known as a very effective thickener of soups and gravies. The gastric emptying of a pHF-W is comparable to that of mother’s milk and significantly faster than that of intact milk protein [56] *L. reuteri* was shown to reduce regurgitation and to enhance gastric emptying [57]. Different studies showed a significant effect on regurgitation within one week of dietary treatment [58,59,60].

Limitations of an open real-life intervention study is that bias and a placebo effect cannot be ruled out. A placebo effect may explain the negative outcome of two studies evaluating the effect of *L. reuteri* DSM 17983 on infantile colic [61,62], while meta-analyses conclude for efficacy of the same strain in colic [53,54]. In future studies, the microbiome composition should be assessed in this kind of intervention trial, and epigenetic effects should be evaluated following formula administration [63,64]. Epigenetics involves several mechanisms including DNA methylation, histone modifications and microRNAs, which can modify the expression of genes [64]. The period between conception, pregnancy and the two first years of life is considered the optimal time for environmental factors, such as nutrition, to exert their beneficial epigenetic effects [64]. Up to now, insufficient attention has been accorded to on how early feeding may have an impact on functional (immunological as well as epigenetic) activity and on establishing epigenetic markers of immunological responses to milk [65]. It would be of interest to prolong the observation period in future studies, in order to evaluate whether these short-term beneficial effects have an ongoing positive effect later in life, and, e.g., decrease irritable bowel syndrome or recurrent abdominal pain in older children.

## 5. Conclusions

These data confirm that reassurance, guidance and nutritional treatment are an effective intervention in the management of FGIDs in infants. A real-life observational study cannot exclude a placebo effect of the dietary intervention, but most important is that symptoms decreased and QoL improved. Since nutritional treatment is safe, it offers the possibility to health care providers to improve the QoL of families with infants with FGIDs without risk of adverse effects.

## Figures and Tables

**Table 1 nutrients-13-03336-t001:** Composition of study formula per 100 mL.

Energy	67 kcal
Lipid (DHA/ARA)	3.4 g (17 mg/17 mg)
Carbohydrates (lactose/starch)	7.5 g (5.0 g/2.2 g)
Fibers (Starch/FOS/GOS)	0.48 g (0.08 g/0.04 g/0.36 g)
Potato starch	2.2 g
Protein (pHF-W)	1.3 g
Magnesium	8.37 mg

Legend: DHA: docosahexaenoic acid; ARA: arachidonic acid; FOS: fructo-oligosaccharide; GOS: galacto-oligosaccharide; pHF-W: whey partial hydrolysate.

**Table 2 nutrients-13-03336-t002:** The Cow’s Milk-associated Symptom Score (CoMiSS^®^) value [12].

Symptom	Score			
Crying	0	≤1 h/day		
	1	1 to 1.5 h/day		
	2	1.5 to 2 h/day		
	3	2 to 3 h/day		
	4	3 to 4 h/day		
	5	4 to 5 h/day		
	6	≥5 h/day		
Regurgitation	0	0 to 2 episodes/day		
	1	≥3 to ≤5 episodes of small volume		
	2	>5 episodes of >1 coffee spoon		
	3	>5 episodes of ± half of the feedings in < half of the feedings		
	4	continuous regurgitations of small volumes >30 min after each feeding		
	5	regurgitation of half to complete volume of a feeding in at least half of the feedings		
	6	regurgitation of the complete volume after each feeding		
Stools (Bristol scale)	4	type 1 and 2 (hard stools)		
	0	type 3 and 4 (normal stools)		
	2	type 5 (soft stool)		
	4	type 6 (liquid stool, if unrelated to infection)		
	6	type 7 (watery stools)		
Skin symptoms	0 to 6	Atopic eczema	Head-neck-trunk	Arms-legs-hands-feet
		Absent	0	0
		Mild	1	1
		Moderate	2	2
		Severe	3	3
	0 to 6	Urticaria (0: no, 6: yes)		
Respiratory symptoms	0	no respiratory symptoms		
	1	slight symptoms		
	2	mild symptoms		
	3	severe symptoms		

**Table 3 nutrients-13-03336-t003:** Patient characteristics.

Number of infants	(boys%/girls%)	196 (55.6/44.4)
Birthweight (gram)	Mean (SD)	3321 (534)
	Median (Q1–Q3)	3320 (3020–3685)
Gestational age (weeks)	Mean (SD)	38.7 (1.6)
	Median (Q1–Q3)	39 (38–40)
Mode of delivery	Vaginal (%)	76.0
	Cesarean section (%)	24.0
Family history of atopic disease	Yes/No (n, %)	63 (32.1%)/132 (67.4%)
At baseline		
Age (months)	Mean (SD)	1.5 (1.0)
	Median (Q1–Q3)	1.1 (0.7–2.1)
Weight (gram)	Mean (SD)	4558 (1120)
	Median (Q1–Q3)	4225 (3780–5175)
Estimated overfeeding	Yes/No (%)	4.6/95.4
Feeding	Intact protein (n, %)	138 (70.4)
	Partial hydrolysate (n, %)	58 (29.6)
FGID	Crying (n, %)	158 (80.6)
	Regurgitation (n, %)	118 (60.2)
	Hard stools (type 1–2) Liquid stools (type 6, 7)	36 (18.8%) 48 (24.5%)

Legend: SD: standard deviation: Q: quartile; FGID: functional gastro-intestinal disorder.

**Table 4 nutrients-13-03336-t004:** The evolution of global CoMiSS.

		Day 3	Day 7	Day 14
N	196	189	182	178
Valid (n; %)	192 (98.0%)	186 (98.4%)	177 (97.3%)	172 (96.6%)
CoMiSS °				
Mean (SD)	6.46 (3.09)	5.21 (2.90)	4.98 (2.93)	4.92 (3.06)
Median (Q1–Q3)	6 (4; 8)	5 (3; 7)	5 (3; 6)	5 (3; 6.5)
Min–Max	0–15	0–16	0–16	0–16
Change in CoMiSS °				
Mean (SD)		−1.29 (3.15)	−1.56 (3.47)	−1.53 (3.68)
*p*		<0.0001 *	<0.0001 *	<0.0001 *
Median (Q1–Q3)		−1 (−3; 1)	−2 (−4; 0)	−1 (−3.5; 7)
Min–Max		−13; 10	−13; 9	−14; 7
Missing n (%)		3 (1.6%)	5 (2.8%)	6 (3.4%)
Min–Max		−24; 19	−31; 24	−18; 25
Missing n (%)		3 (1.6%)	5 (2.8%)	7 (3.9%)

Legend. n: number; °: change is related to baseline; SD: standard deviation; Q: quartile; *p*-value * Wilcoxon test for skewed variables; Min: minimal; Max: maximal.

**Table 5 nutrients-13-03336-t005:** Evolution of specific CoMiSS per time point.

Symptoms	Baseline	Day 3	Day 7	Day 14
n	196	189	182	178
Crying	2.24	1.72	1.29	1.23
Regurgitation	1.31	0.76	0.75	0.72
Stools °	2.53	2.44	2.64	2.59
Skin symptoms	0.20	0.13	0.10	0.15
Respiratory symptoms	0.18	0.17	0.20	0.25
Total CoMiSS	6.46	5.21	4.98	4.92

Legend: n: number; °: CoMiSS uses the Bristol stool scale to describe stool consistency; both liquid and hard stools score high.

**Table 6 nutrients-13-03336-t006:** Evolution of CoMiSS per time point for stool consistency.

Stool Type	Baseline	Day 3	Day 7	Day 14
n	196	189	182	178
1 and 2 (Hard)	36 (18.8%)	12 (6.5%)	10 (5.7%)	4 (2.2%)
3 and 4 (Normal)	40 (20.8%)	41 (22.0%)	34 (19.2%)	31 (17.4%)
5 (Soft)	68 (35.4%)	69 (37.1%)	58 (32.8%)	65 (36.5%)
6 (Fluid)	41 (21.4%)	58 (31.2%)	69 (39.0%)	66 (37.1%)
3–6 (nl for infants)	149 (76.0%)	168 (88.9%)	162 (89.0%)	162 (91.0%)
7 (Watery)	7 (3.7%)	6 (3.2%)	6 (3.3%)	6 (3.4%)
Missing	4 (2.0%)	3 (1.6%)	5 (2.7%)	6 (3.4%)

Legend: n: number; nl: normal.

**Table 7 nutrients-13-03336-t007:** The evolution of CoMiSS in the subgroup with CoMiSS > 9.

		Day 3	Day 7	Day 14
N				
Valid (n; %)	43	42 (97.7%)	40 (97.6%)	38 (95.0%)
Change in CoMiSS from baseline				
Mean (SD)		−3.86 (3.22)	−4.40 (3.69)	−4.55 (4.58)
*p*		<0.0001 *	<0.0001 *	<0.0001 *
Median (Q1–Q3)		−4 (−6; −2)	−5 (−6; −2)	−5 (−8; −1)
Min–Max		−13; 4	−13; 3	−14; 5
Missing n (%)		3 (1.6%)	5 (2.8%)	6 (3.4%)
Min–Max		−24; 19	−31; 24	−18; 25
Missing n (%)		1 (2.3%)	1 (2.4%)	2 (5.0%)
Total CoMiSS	10.73	6.93	6.43	6.11
Crying	3.43	2.31	1.73	1.97
Regurgitation	2.73	1.43	1.40	1.16
Stools	3.77	2.76	2.85	2.42
Skin + resp symptoms	0.80	0.43	0.46	0.56

Legend: n: number; SD: standard deviation; Q: quartile; *p*-value * Student’s paired *t*-test for normally distributed variables; Min: minimal; Max: maximal; resp: respiratory.

**Table 8 nutrients-13-03336-t008:** Evolution of the QUALIN score per time point.

QUALIN	Baseline	Day 3	Day 7	Day 14
n	191	186	177	171
Mean (SD)	22.7 (9.1)	23.6 (8.5)	24.0 (9.3)	25.8 (8.5)
Median (Q1–Q3)	22 (18; 27)	24 (16; 30)	23 (18; 30)	25 (19; 32)
Min–Max	(−1; 63)	(3; 42)	(−8; 52)	(8; 50)
Missing	5 (2.6%)	3 (1.6%)	5 (2.8%)	7 (3.9%)
Change from baseline		1.0 (7.0)	1.4 (7.8)	3.2 (8.2)
*p*		0.065 *	0.008 **	<0.0001 *
QUALIN topic				
Behavior/communication	7.27	8.34	8.90	10.33
Ability to remain alone	3.58	3.24	3.26	3.10
Family environment	6.20	6.48	6.70	6.93
Psychological and somatic well-being	−2.19	−3.06	−3.89	−3.93
Other	7.85	8.58	9.02	9.34
Total QUALIN score	22.71	23.58	23.99	25.77
Total QUALIN score in subgroup with CoMiSS > 9	22.48	23.33	25.30	26.27

Legend: n: number; SD: standard deviation; Q: quartile; Min: minimal; Max: maximal. * Student’s paired *t*-test for normally distributed variables; ** Wilcoxon test for skewed variables.

## Data Availability

No extra data available.

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
