# Peer review of "An Observational Real-Life Study with a New Infant Formula in Infants with Functional Gastro-Intestinal Disorders"

_nutrients, 2021, doi:10.3390/nu13103336_

Round 1
Reviewer 1 Report
In the present paper (An observational real-life study with a new infant formula in infants with functional gastro-intestinal disorders), Vandenplas and coworkers examined the efficacy of a formula with a partial whey hydrolysate, starch, high magnesium content, prebiotic fructo-oligosaccharide and galacto-oligosaccharide and the probiotic Lactobacillus reuteri DSM 17938 during two weeks in 196 infants with at least two functional gastro-intestinal disorders (FGIDs). The authors concluded that nutritional intervention was shown to be effective in infants with FGIDS in real life circumstances. Overall, I think that the manuscript is well-written (within the scope of this journal), well-structured and the data are of clinical relevance.
I would like to make some suggestions on how to make the paper stronger.
Based on these interesting data, it is possible predict if this formula, can cause long-term effects (both side effects and/or favourable effects) in adults?
Recent research suggested that epigenetic effects were associated with the gut microbiome composition. Does the authors plan to assess microbioma composition in their intervention trials or to investigate epigenetic effects following formula administration? Please make a comment in the discussion section of revised manuscript.
The authors could add in Graphical form the molecular targets of treatment; in this way, I feel that the readers can better understand this crucial topic, then supporting those arguments.
Author Response
Dear Reviewer
We thank you for your positive comments and suggestions. Please find below the changes in response to your suggestions.
In the present paper (An observational real-life study with a new infant formula in infants with functional gastro-intestinal disorders), Vandenplas and coworkers examined the efficacy of a formula with a partial whey hydrolysate, starch, high magnesium content, prebiotic fructo-oligosaccharide and galacto-oligosaccharide and the probiotic Lactobacillus reuteri DSM 17938 during two weeks in 196 infants with at least two functional gastro-intestinal disorders (FGIDs). The authors concluded that nutritional intervention was shown to be effective in infants with FGIDS in real life circumstances. Overall, I think that the manuscript is well-written (within the scope of this journal), well-structured and the data are of clinical relevance. |
We thank the reviewer for the positive comment |
Based on these interesting data, it is possible predict if this formula, can cause long-term effects (both side effects and/or favourable effects) in adults? |
We thank the reviewer for this interesting suggestion. We adapted the manuscript as suggested: "It would be of interest to prolong the observation period in future studies, in order to evaluate if these short term beneficial effects have an ongoing positive effect later in life, and e.g. decrease irritable bowel syndrome or recurrent abdominal pain in older children." |
Recent research suggested that epigenetic effects were associated with the gut microbiome composition. Does the authors plan to assess microbioma composition in their intervention trials or to investigate epigenetic effects following formula administration? Please make a comment in the discussion section of revised manuscript. |
The reviewer makes again a very interesting suggestion. The microbiome composition was not determined in this study. We added this as a limitation. |
The authors could add in Graphical form the molecular targets of treatment; in this way, I feel that the readers can better understand this crucial topic, then supporting those arguments.
|
We do think that the observational study does not allow to go deeper in the "molecular targets" |

Reviewer 2 Report
With interest, I read the manuscript nutrients-1370336. It reports the results of a valuable observational trial study in the field of nutritional/food science.
Comments:
1. Usage of CoMiSSTM should be (better) explained already in the Methods, i.e. why this cow’s milk-related scoring system was used in the present study. There is some explanation later, in the Discussion (starting lines 168-173) but it is too late and 1-2 sentences should be given already in the Methods.
2. Statistics requires several clarifications:
a. “A two-sided 95% confidence interval (CI) was considered as a default.“. What do you mean here exactly? Can we see those CIs somewhere?
b. How was the normality oft he distribution assessed?
c. “ANOVA and Student t-test were used to compare means …“. Actually, this is much too much simplified (“means“?). Either write it more in detail or just write “to compare independent continous variables“ or something like that.
d. Any post-hoc tests after ANOVA?
e. What unpaired comparison tests were used if the distribution was different than normal?
f. “Stepwise multivariable analyses“ or “stepwise multivariate analyses“ (PMID: 23153131)? Those you use interchangeably.
g. In general, please, characterize your regression model(-s) more in detail.
3. This work is clinical but a few mechanistic sentences on the functional effects of the bacteria and oligosaccharides present in your formula as summarized in some recent relevant reviews (PMID: 33193294 and 33668787) should be added in the Discussion.
Author Response
Dear Reviewer
We thank you for your constructive comments and suggestions. Please find our response to your comments below.
With interest, I read the manuscript nutrients-1370336. It reports the results of a valuable observational trial study in the field of nutritional/food science. |
Thank you. |
Usage of CoMiSSTM should be (better) explained already in the Methods, i.e. why this cow’s milk-related scoring system was used in the present study. There is some explanation later, in the Discussion (starting lines 168-173) but it is too late and 1-2 sentences should be given already in the Methods. |
We did add the following in the Methods: The CoMiSS was developed as a symptom score to quantify the symptoms in a trial com-paring the efficacy of two extensive hydrolysates in infants suspected to suffer cow's milk protein allergy (CMPA) [13,14]. |
2. Statistics requires several clarifications: a. “A two-sided 95% confidence interval (CI) was considered as a default.“. What do you mean here exactly? Can we see those CIs somewhere? |
We did not derive the 95% confidence intervals for the descriptive statistics. Instead, we have shown the p-values, standard deviation and interquartile range.
The 95% confidence intervals were shown for the multivariate regression analyses. These are displayed in Table 1 and Table 2 in the Supplementary Materials.
We adjusted the Methods by adding the following sentences: “Statistical significance was set at a level of 0.05. Standard deviation and inter-quartile ranges are reported for descriptive statistics.” […] “Standard errors and two-sided 95% confidence intervals (CIs) are reported for the covariate estimates of the multivariate analyses (see Supplementary Materials.” |
How was the normality of the distribution assessed? |
We added the following sentence in the Methods:
“The normality of the QUALIN and CoMiSS scores was assessed through the Shapiro Wilk Test. Normality was assumed if the p-value of the test statistic was larger than the significance level of 5%.” We also indicated this in the footnote of Table 4, 7 and 8 through an asterisk (*) or (**). |
ANOVA and Student t-test were used to compare means …“. Actually, this is much too much simplified (“means“?). Either write it more in detail or just write “to compare independent continuous variables“ or something like that. |
We adapted the following sentence in the Methods: “ANOVA and Student t-test were used to compare the means of independent continuous variables. |
Any post-hoc tests after ANOVA? |
No ANOVA post-hoc tests were performed because only two groups were compared: scores at baseline versus scores at day 3, day 7 or day 14.
The following other subgroup analyses were performed: 1) the subgroup analysis of CoMiSS score > 9 at baseline; 2) a re-analysis of the stool index focusing on changes in Types 1-2; 3) subgroup analysis excluding patients who received gastro-intestinal medications during the study |
What unpaired comparison tests were used if the distribution was different than normal? |
We did not perform unpaired comparison tests since the data was paired. For paired, non-normal data we used the Wilcoxon test instead of the Student t-test. This was indicated in Table 4, 7 and 8 through an asterisk (*) or (**).
|
“Stepwise multivariable analyses“ or “stepwise multivariate analyses“ (PMID: 23153131)? Those you use interchangeably. |
Thank you for noticing. We changed the wording everywhere into “stepwise multivariate analyses” to be consistent. |
In general, please, characterize your regression model(-s) more in detail. |
We added in the Methods extra details on the regression models. We also specified the procedure that was used in SAS (i.e. proc glimmix). The following paragraph was added in the Methods:
“Multivariate mixed effect model analyses were planned to assess change from baseline score over time (repeated measures). Multivariate mixed effect model was accommodated within-subject (patient in this case) nonindependence or within-unit clustering (physician in this case) and unobserved heterogeneity, at once and allowed for subject-specific conclusions. This was controlled by introducing a random effect (patient) in the model (Schober et al, 2018). Time (visit day 1, home report day 3, home report day 7, visit day 14) was also included as a discrete variable and fixed effect since the main interest was the change in mean scores (Lee et al, 2019). Other covariates or factors (i.e. age, sex, BMI, gestational age, birth BMI, presence of adverse events, history of allergy, mode of delivery, number of siblings, baseline CoMiSS, baseline QUALIN, type of FGID at baseline) were also included in the model as fixed effects. The inclusion of baseline score as covariate can help to decrease variability in random errors and increase the probability of detecting the significant time effect on the mean change from baseline in score (Lee et al, 2019). Different models were constructed starting from the full model from which covariates are removed sequentially. The selection of the best model was based on the Akaike Information Criterion (AIC). The model with the lowest AIC was selected for the multivariate analysis. The normality of the QUALIN and CoMiSS scores was visually assessed through frequency distributions showing the spread of the scores. Normality was assumed if the distribution was bell-shaped and symmetric around the mean. Quantile-quantile (Q-Q) plots were constructed to assess if the choice of the distribution was acceptable. Dots perfectly aligned with the straight line mean perfect choice of distribution. The “proc glimmix” procedure in SAS was used for the multivariate mixed effect model analyses in which a Gamma distribution and log link function was selected to model non-normal responses, otherwise a normal distribution was specified. Standard errors and two-sided 95% confidence intervals (CIs) are reported for the covariate estimates of the multivariate analyses (see Supplementary Materials).”
|
This work is clinical but a few mechanistic sentences on the functional effects of the bacteria and oligosaccharides present in your formula as summarized in some recent relevant reviews (PMID: 33193294 and 33668787) should be added in the Discussion. |
We added both references in the discussion section. Addressed in the discussion. |

Round 2
Reviewer 2 Report
Thank you for addressing my comments well. I have no further remarks.